# Magnetoencephalography: Clinical and Research Practices

**DOI:** 10.3390/brainsci8080157

**Published:** 2018-08-17

**Authors:** Jennifer R. Stapleton-Kotloski, Robert J. Kotloski, Gautam Popli, Dwayne W. Godwin

**Affiliations:** 1Department of Neurology, Wake Forest School of Medicine, Winston-Salem, NC 27101, USA; gpopli@wakehealth.edu (G.P.); dgodwin@wakehealth.edu (D.W.G.); 2Research and Education, W. G. “Bill” Hefner Salisbury VAMC, Salisbury, NC 28144, USA; 3Department of Neurology, William S Middleton Veterans Memorial Hospital, Madison, WI 53705, USA; kotloski@neurology.wisc.edu; 4Department of Neurology, University of Wisconsin School of Medicine and Public Health, Madison, WI 53726, USA; 5Department of Neurobiology and Anatomy, Wake Forest School of Medicine, Winston-Salem, NC 27101, USA

**Keywords:** magnetoencephalography, magnetic source imaging, synthetic aperture magnetometry, epilepsy

## Abstract

Magnetoencephalography (MEG) is a neurophysiological technique that detects the magnetic fields associated with brain activity. Synthetic aperture magnetometry (SAM), a MEG magnetic source imaging technique, can be used to construct both detailed maps of global brain activity as well as virtual electrode signals, which provide information that is similar to invasive electrode recordings. This innovative approach has demonstrated utility in both clinical and research settings. For individuals with epilepsy, MEG provides valuable, nonredundant information. MEG accurately localizes the irritative zone associated with interictal spikes, often detecting epileptiform activity other methods cannot, and may give localizing information when other methods fail. These capabilities potentially greatly increase the population eligible for epilepsy surgery and improve planning for those undergoing surgery. MEG methods can be readily adapted to research settings, allowing noninvasive assessment of whole brain neurophysiological activity, with a theoretical spatial range down to submillimeter voxels, and in both humans and nonhuman primates. The combination of clinical and research activities with MEG offers a unique opportunity to advance translational research from bench to bedside and back.

## 1. Introduction

### 1.1. What Is MEG?

The fundamental activity of the brain, signaling between neurons, functions through the movement of charged ions. The resultant fields generated from these charges, i.e., electrical potentials, are often measured for clinical and research purposes, mostly commonly through electroencephalography (EEG). Moreover, changing electric fields generate magnetic fields which also are measured clinically through magnetoencephalography (MEG), and the latter has several advantages over the measurements of electric fields, although MEG also requires greater technical infrastructure (see [1] for review). While electric neurophysiologic activity has been measured since 1875 [2], the technical advances required to enable the measurement of magnetic neurophysiologic activity were not available until nearly a century later [3]. While EEG measures the secondary extracellular electrical potentials generated by brain activity, MEG detects the magnetic fields associated with the primary, or intracellular activity. As the brain, skull, and scalp distort the EEG signals, EEG only has a minimum spatial resolution of 2 to 3 cm [4]. In contrast, the head is magnetically transparent, so under optimal conditions MEG activity maps have a sub-millimeter spatial accuracy [5,6]. MEG can also be used to image subcortical brain activity in such areas as the hippocampus, amygdala, thalamus, and brainstem [7,8,9,10,11,12,13,14,15,16,17], as well as cerebellum [9,18,19], although this ability is often not appreciated.

MEG is a quantitative neurophysiological recording technique that can be used to perform neuroimaging. The analytical process of localizing or mapping MEG signals within the brain is referred to as magnetic source imaging (MSI). Typically, MSI is accomplished by dipole analysis, in which a brief signal, such as an interictal spike, is modeled as a point source arising from a single generator [1]. Dipole brain maps of interictal spikes look similar to clusters of pins embedded in a map (e.g., see analysis for patient in Section 3.4). The synthetic aperture magnetometry (SAM) beamformer is an alternative MSI technique to dipole analyses that does not make assumptions regarding the durations, numbers, locations, or spatial extent of generators present [20,21,22,23,24]. Through the use of SAM [20], MEG signals can be converted into statistical parametric maps of brain activity, an mapping approach similar to those obtained by functional magnetic resonance imaging (fMRI) to map the blood oxygen-level dependent (BOLD) signal [5], and which (in the case of MEG) provides much more detailed spatial information than that obtained by dipole analyses.

fMRI is an indirect measure of brain activity based on blood oxygen content, thus fMRI signals are very slow, with a minimum temporal resolution of ≥2 s. In contrast, MEG signals are a direct measure of the primary currents of brain activity [1], have a sub-millisecond temporal resolution [25], and can have a frequency bandwidth of direct current (DC) to 3000 Hz (or greater), which is far beyond what fMRI can detect and which has particular importance in the detection of high frequency oscillations, or HFOs (see below). Finally, SAM can be used to construct virtual electrodes for any point in the brain, providing a continuous, wide-band, submillisecond representation of brain activity. Virtual electrodes essentially provide similar neurophysiological information as an invasive intracranial electrode [21,26], with the obvious benefits of not causing tissue damage, the ability to examine locations retrospectively, superior spatial resolution, and the ability to target deep structures such as the thalamus [9], brainstem [27], or cerebellum [19], which are dangerous to access with implantable electrodes.

### 1.2. Clinical Uses for MEG

Epilepsy is a devastating disorder, defined by seizures emanating from a brain predisposed to generate seizures, which be inherent (i.e., genetic) or acquired through a precipitating event, such as a traumatic brain injury [28]. Epilepsy is the fourth most common neurological disorder in the United States. Seizures are the result of abnormal, synchronous firing of populations of neurons that lead to a behavioral change, including behavioral arrest, uncontrolled movement, and/or loss of consciousness [29]. Approximately 1.8% of American adults have been diagnosed with epilepsy at some point in their lives, about 1.0% of all American adults are classified as having active epilepsy [30], and the prevalence of epilepsy in children is about 0.5–1% [31,32]. Of these individuals, only about 70–80% achieve adequate seizure control through the use of antiseizure drugs [33]. The remaining 20–30%, whose seizures are pharmaco-resistant, are responsible for ~80% of the estimated $12.5 billion annual expenditure on epilepsy [34]. For this latter group, surgical resection of the seizure onset zone(s) is offered as an effective means for ameliorating or alleviating epilepsy [35], although it is estimated that only 2–3% of all possible surgical candidates actually receive surgery [36]. In both children and adults, successful surgery is linked to improved psychiatric and social function, improved quality of life, and especially in the case of children, improved development (see [37,38] for review).

To better distinguish between individuals who are surgical candidates and those who are not, and to increase the probability of successful surgery in the former group, it is necessary first to localize the seizure generators. While the gold standard for localizing the seizure onset zone is the clinical outcome following resection, during a presurgical evaluation intracranial EEG (ICEEG) is considered to be the most reliable technique. However, ICEEG is limited as it can only identify the locations of the ictal generator(s) if the electrode grids, strips, or depths are placed directly over or within the seizure focus, and only a very limited portion of the brain can be monitored with ICEEG. Therefore, accurate and precise data regarding the seizure onset zone are needed prior to the implantation for an optimal investigation. Furthermore, ICEEG is an expensive and invasive procedure [39], requiring lengthy inpatient monitoring over days or weeks, and carries the rare but serious risks of infection, bleeding, and scarring. As such, a variety of supplemental, noninvasive techniques such as scalp EEG, MRI, positron emission tomography (PET), single-photon emission computed tomography (SPECT), and MEG/MSI are commonly used to localize the seizure onset zone or the related irritative zone, and to direct the placement of ICEEG.

For over thirty years [40,41,42,43], MEG and MSI have been used extensively to accurately localize the irritative zone in individuals with epilepsy. The average sensitivity of MEG to detect epileptiform activity is ~70–80%, versus about a 60% detection rate for simultaneously recorded EEG [44,45,46], while in those who proceeded to surgery, MSI accurately identified the lobe to be treated with 89% accuracy [44]. In ~13% of all individuals, MEG-only spikes are detected, while in ~3% of all individuals EEG-only spikes occur, suggesting that in one-third of all EEG-negative individuals, MEG is expected to detect epileptiform signals [45]. Numerous clinical studies have demonstrated an excellent agreement between foci delineated by MEG/MSI and ICEEG [26,47,48,49,50,51,52], and MEG foci have been found to align with lesions such as tuberous sclerosis [53,54,55,56,57,58] and cortical dysplasia [50,51]. Similarly, retrospective studies have found that dipole maps of interictal spikes had clustered over subsequent resection volumes [46,59,60], even when video EEG results were nonlocalizing. In addition to spikes, HFOs (~100–500 Hz, and even up to or beyond 1000 Hz [53,61]) are also linked to the seizure onset zone, can occur independently of spikes, and represent an additional biomarker of primary epileptogenesis [62,63,64,65]. Complete resection of tissue related to ictal HFOs is also associated with good surgical outcome [66,67,68]. While HFOs are typically recorded with ICEEG, they may also be detected noninvasively by MEG [61,69,70,71,72]. Finally, MEG and MSI can also be used to map eloquent cortex as a part of surgical planning, another task that is often performed invasively with ICEEG. MSI has been used to map such functional regions as somatosensory [73,74,75,76,77,78,79,80,81,82,83,84,85], motor [86,87,88,89], visual [90] and language networks [91,92,93,94,95,96]. 

Several lines of evidence directly support the utility of MEG as a key, nonredundant part of the surgical planning routine. In a prospective, blinded, crossover study, Sutherling and colleagues [97] compared surgical planning decisions before and after MSI findings were revealed. They found that MSI provided useful, nonredundant information in 33% of all patients, either by changing the surgical decision or by adding ICEEG electrodes or by changing ICEEG coverage. MSI benefitted 21% of patients that proceeded to surgery and added useful information that changed treatment in 9% of patients, without increasing complications. These results have been corroborated by Mamelak and colleagues, who found that MSI provided unique localizing information not revealed by other methods, that MSI changed ICEEG placement, and that MSI strongly influenced surgical management, particularly in patients with neocortical epilepsy [48]. In another series of landmark studies, Knowlton and colleagues [98] found that MSI has a higher sensitivity (58–64%) and specificity (79–88%) than PET (sensitivity, 22–40%; specificity, 53–63%) or ictal SPECT (sensitivity, 39–48%; specificity, 44–50%) when compared to ICEEG-based localization; the authors note that these estimates may be low because ICEEG itself has limitations in delineating foci. MSI can guide ICEEG placement and thereby increase its diagnostic yield [39,48], or it might be able to reduce the use of ICEEG [52]. Positive MSI results also have 72% sensitivity, 70% specificity, 78% positive predictive value, and 64% negative predictive value with respect to a subsequent freedom from disabling seizures (Engel class I surgical outcome) in patients who require ICEEG [99].

MEG is less commonly utilized in pediatric patients with epilepsy owing to their smaller head sizes and the corresponding increased distance from the sensors as well as difficulties in cooperation, but children have been successfully scanned in both adult, whole cortex systems [50,56,100] as well as helmets and arrays specially designed for children [101,102,103,104,105,106,107]. Children may also be sleep deprived prior to the scan, thus encouraging them to sleep (and remain motionless) in the MEG [53,102], or they may be sedated [50,56]. As with adults, MEG is useful in both lesional and nonlesional cases [38,101], with particular utility in mapping spikes associated with tuberous sclerosis [53,55,56,57,102], cortical dysplasia [102,108] and other malformations [101,108]. HFOs have been successfully localized by MEG in children [71] and functional mapping of eloquent cortex may also be performed [50,102]. 

The American Clinical Magnetoencephalography Society (ACMEGS) is a clinical society comprised of physicians and researchers with expertise in MEG/MSI, EEG, MRI, and CT. Based on the past several decades of MEG research, in 2009 the ACMEGS group released their position statement in which they support the routine use of MEG/MSI in the presurgical evaluation of epilepsy for the following reasons: (1) It can safely and cost-effectively provide localizing information in comparison to invasive procedures; (2) It can increase the yield of ICEEG by identifying or refining areas for coverage; and (3) It may reduce costs and improve the accuracy of epilepsy evaluations, and thus make surgery a more appealing option [25].

## 2. MEG at Wake Forest Baptist Health

As an example of a MEG program that incorporates both active clinical and research programs, we will describe our own work at Wake Forest Baptist Health (WFBH). WFBH is an academic medical center located in Winston-Salem, NC, USA. WFBH possesses a whole helmet, 304-channel CTF MEG International Services LP MEG scanner (Coquitlam, BC, Canada), which was installed in early 2006. While there are about 30 clinical MEG sites located in the United States, regionally, our installation is the only one in North Carolina, and the only clinical scanner between Birmingham, AL, and the National Institutes of Health (NIH) campus (Bethesda, MD, USA) (an area with an approximately 300 miles/480 km radius). As such, the service receives referrals from within state as well as throughout the mid-Atlantic States and has even received patients from as far as New York, Alabama, and Missouri. Unlike many other sites that utilize planar gradiometers, our CTF MEG system is equipped with axial (also called radial) gradiometers, which have an excellent depth profile [109,110] and can thus effectively detect subcortical signals [7,8,10,11,12,13,14]. The CTF MEG system also employs synthetic gradiometry, which very effectively removes environmental noise from the patient’s magnetoencephalogram while boosting the signal to noise ratio (SNR) and results in remarkably clean, stable data [22,78,109,110,111].

For MSI, we utilize synthetic aperture magnetometry (SAM) to identify the irritative zone and to map eloquent brain functions. Unlike dipole analyses, SAM can easily map multiple interictal generators [21,23,24], identify generators of large spatial extent [22], and properly estimate the depth of signals [20]. As SAM is a spatial filter which automatically removes all signals that do not arise from within the brain volume, potential artifacts from heartbeat, respiration, eye movements [87], very large amplitude artifacts arising from dental hardware [80,112], deep brain stimulators [27], etc. are reliably eliminated. One variant of this technique, SAM(g2), has been used successfully in the clinical setting to map the kurtotic (sharp) signature of epileptiform signals arising from interictal foci and to reconstruct the virtual electrodes associated with these generators [21,24,51,53,54,112,113,114]. Importantly, the virtual electrodes can be inspected for the presence of epileptiform activity, and exhibit a strong agreement with the activity recorded by ICEEG [21,26]. A second variant of the SAM method, dual-state SAM, can be used to image trial-based tasks typically employed for functional mapping [87,91,93]. We have recently extended the uses of SAM to successfully map patients with vagal nerve stimulators (VNS, [11]; see below for examples), cerebrospinal fluid (CSF) shunts, metal plates, and cardiac pacemakers. 

### General MEG Scanning Protocols

MEG recordings can be obtained in either the awake state or in a sedated state. There are no age or sex exclusions to receive a MEG scan at our facility, though individuals with a head circumference >63 cm will not fit within the helmet and therefore cannot receive a MEG scan. Those with a cochlear implant are excluded from scanning because CTF has informed us that signals from the implants may damage our gradiometers. We do not have any other restrictions for obtaining a clinical MEG scan. The CPT codes for MEG scans of epileptiform activity and cognitive function are 95965, 95966, 95967, and the code for magnetic source imaging is S8035.

Typically, a clinical MEG scan for epilepsy has a total duration of 40–60 min, and this period is broken into five or ten minute epochs for better file size management on our acquisition system. EEG is usually acquired simultaneously with the MEG scan if the individual’s head circumference will permit the addition of EEG leads. Our system can perform continuous head localization to track head position and movement in the scanner through the use of three small, energized fiducial leads that are placed between the eyebrows and in front of each ear. The total time to prepare the individual and to attach the fiducials and the EEG electrodes is about one hour. The MEG scanner operation is silent, and the helmet reclines, so individuals can be scanned supine while sleeping. Adults are usually sleep-deprived overnight, or for several hours for children, because interictal spikes may occur more frequently during sleep [115]. Following the epilepsy scan portion, the EEG leads are removed and the patients are given the option of a short break. Scanning recommences in the seated position, and patients may undergo somatosensory, motor, or language testing for mapping purposes, as per physician request. The functional testing portion typically requires a total of about an hour. The total setup and scan time for a MEG is about 3–4 h. Following the MEG scan, individuals receive a structural T1 magnetization-prepared rapid gradient-echo (MP-RAGE) MRI to subsequently align with their MEG data, and from which a realistic three-spherical shell, multiple local-spheres head model will be constructed [116]. Patients also receive additional imaging such as DTI (diffusion tensor imaging), T2 FLAIRs (fluid-attenuated inversion recovery), SWI (susceptibility weighted imaging), FS (fat saturation), and IR (inversion recovery) scans. The MP-RAGE and FLAIRs are conducted before and after gadolinium contrast. The MRI scans usually occur in the same day. For the purposes of MEG analysis, only the T1 MPRAGE is used. To prevent subtle magnetization effects, MRI scans are always performed after MEG scans.

MEG data are acquired with a sampling rate of 600–2400 Hz, and data are preprocessed offline with synthetic 3rd order gradiometry, DC-offsetting, and powerline filtering [11,117,118,119]. The SAM(g2) method is comprised of a sequence of steps in which the sensor data are filtered from 20–70 Hz and then beamformed in single-state mode. Voxels with a pseudo-z score ≥ 5.0 are retained and virtual electrodes of 20–70 Hz bandwidth are constructed for each voxel. The kurtosis (g2) of each virtual electrode is calculated and then a map of g2 scores is created. Spikes are detected as outliers (kurtotic) in this bandwidth. Voxels for which g2 ≥ 3.0 are retained, and the local maxima plus the full width, half max (FWHM) volume of voxels surrounding the peaks in the map correspond to the location(s) of interictal spikes, which are mapped in brain space. Virtual electrodes are constructed for all peak voxels in a bandwidth of 3–70 Hz. Spikes are automatically marked in the virtual electrodes at the earliest rising phase of all waveforms with a peak-to-rms ratio ≥ 6.0 [23,112], although spikes with a lower ratio are often visible and may be considered. Spikes in the virtual electrodes may then be compared to the simultaneous EEG by the physicians. 

## 3. Illustrative Clinical Cases

Since 2006, MEG and MSI have been part of the standard WFBH presurgical evaluation for patients with epilepsy. We incorporated SAM into our analysis routines in 2010, and our unique combination of axial gradiometers, synthetic gradiometry, and SAM provides an excellent level of accuracy and resolution for epilepsy and functional mapping. We present a small number of representative cases to illustrate the capability of MEG for patient care. When available, we will benchmark the MEG results against the results obtained from other modalities such as scalp EEG, structural MRI, PET, SPECT, and ICEEG. The patients presented in cases 1 and 5 were also presented previously [11].

### 3.1. Case 1: MEG Localization Is Concordant with Multiple Other Modalities

A 37-year-old man with a VNS presented for further evaluation. The patient had focal seizures with dyscognitive symptoms and occasionally evolution to bilateral convulsive seizures since he was 18 years old. Several events were captured during an epilepsy monitoring unit (EMU) admission, all of which localized to the right temporal region. An MRI of his brain did not reveal any structural abnormalities. SPECT (Figure 1F) and PET imaging (Figure 1G) suggested a right temporal seizure focus. A MEG recording analyzed by SAM(g2) revealed a right mesial temporal focus centered on the hippocampus (Figure 1B) and amygdala (Figure 1C), with some occasional right ventral frontal, right posterior, and lateral temporal spread from this zone, and some additional spread to right insula. While the raw MEG sensor data exhibited artifacts due the VNS, the virtual electrode data from hippocampus and amygdala (Figure 1A) displayed no evidence of VNS artifact. While clear spikes existed in the virtual electrode data, the EEG data only occasionally exhibited simultaneous interictal spikes. Invasive monitoring was planned on the basis of the concordant findings between MEG, SPECT, and PET. Subdural grids were placed over the lateral and mesial aspects of the right temporal lobe, and depth electrodes were inserted into the right amygdala and right anterior and posterior hippocampus. Frequent interictal spikes were seen on the hippocampal and amygdalar electrode contacts (Figure 1D), and several seizures arose from the anterior hippocampal and amygdalar electrodes, an example of which can be seen in Figure 1E. Following the invasive monitoring the patient received a right anterior temporal lobectomy with amygdalohippocampectomy. Prior to his surgery, the patient experienced ~2 seizures per month. Following surgery, the patient was seizure-free for several months, but experienced a breakthrough of two seizures following a dose reduction in antiseizure medication, and another breakthrough of four seizures coincident with the onset of an illness. 

### 3.2. Case 2: MEG Discrimates among Multiple Seizure Foci

A 24-year-old man whose focal seizures began at age 7 presented for further evaluation. During his episodes, the patient was reported as having a surprised look on his face, covering his mouth with his hands, and laughing. These episodes happened several times per day and were followed by marching movements of the legs and grunting noises. The patient had no alteration of consciousness during his seizures. He also experienced hypermotor seizures arising out of sleep at least nightly and often had several per night. The patient sustained bilateral frontal lobe damage, intraparenchymal hemorrhages, and extra axial hemorrhages following a severe fall in 2006. He was seen by several neurologists before coming to WFBH. 

The patient’s epilepsy was refractory to multiple antiseizure medications. His initial EMU admission in 2001 was unable to localize the seizure onset zone. The interictal EEG showed very frequent epileptiform discharges arising from the right frontal region, but PET showed mild decreased activity in the left medial temporal lobe which was consistent with an epileptogenic focus. Both ictal and interictal SPECT indicated slightly asymmetric activity within the temporal lobes with the right side greater than the left, also suggestive of an epileptogenic focus within the left temporal lobe. Because of the discordant findings, the patient proceeded to invasive monitoring. ICEEG pointed towards a right frontal lobe origin, but the pattern of spiking occurred almost simultaneously with patient’s clinical semiology. At that time, it was felt that the seizure focus was not clearly localized to warrant right frontal lobectomy. The patient then received a VNS in addition to medication, and this combination was effective for several years.

The patient began to experience more problems around 2012 and was recommended for a MEG scan. The patient’s MEG data were very noisy owing to metal hardware in his skull that covered the burr holes from his previous invasive monitoring, but SAM(g2) sufficiently removed these artifacts and revealed a single focus in the right middle frontal gyrus (Figure 2A) with numerous, MEG-only spikes (Figure 2B). (Because the head is magnetically transparent, MEG is not susceptible to breach effects from craniotomies or other defects.) The MRI obtained for the MEG scan also revealed a small focus of cortical thinning and irregularity with subtle T2 hypointensity in the right frontal lobe. Based on the MEG results, the patient underwent invasive monitoring with a stereo-EEG array placed over the MEG focus (Figure 2C). The patient’s epileptiform activity (green arrowhead, Figure 2D) was subsequently localized to the three ICEEG contacts (green arrows) nearest to the peak of the SAM focus (red sphere, Figure 2C). The patient then received a right frontal lobe resection of the seizure focus and has been seizure free since. 

### 3.3. Case 3: MEG Identifies an Unexpected Seizure Focus

A 21-year-old man with an onset of seizures in 2004 presented for further evaluation. He had one febrile seizure in infancy and has a history of headaches. His events consisted of staring off for a few minutes, being unaware of surroundings, and exhibiting abnormal behavior and incomprehensible speech. He felt tired afterwards and typically had an aura of a right frontal headache. The patient’s seizure frequency was 1–2 per week, and his longest seizure-free interval was two years. The patient had tried multiple antiseizure medications but his seizures were insufficiently controlled under them. During a subsequent EMU admission, scalp EEG recorded several seizures originating in the left temporal region, and his interictal EEG was notable for occasional left temporal slowing and left temporal epileptiform activity. Ictal SPECT exhibited left occipital and left temporal hyperperfusion during one of these seizures, and a later interictal SPECT uncovered two possible anterior and posterior left temporal foci that were considered worrisome for seizure foci. The PET scan was negative. Structural MRI indicated a possible subtle form of hippocampal malrotation as well as two small foci of T2 hyperintensity in the bilateral frontal white matter that were nonspecific, but were thought to be the sequelae of prior ischemia, inflammation/infection, trauma or demyelination. In contrast, the patient’s MEG scan revealed a left occipital focus (Figure 3A) with hundreds of MEG-only spikes visible on the virtual electrode (Figure 3B). Based on the collective results, the patient received invasive monitoring with inter-hemispheric and left lateral occipital grids and left temporal depth electrodes. ICEEG confirmed the left occipital MEG focus as the seizure generator. The patient had a left occipital resection and had a few seizures shortly after surgery in the setting of medication nonadherence. Following this, he has been seizure-free for over a year.

### 3.4. Case 4: SAM Localization Is Superior to Dipole Analysis

A 31-year-old woman with a VNS implantation three years prior to her MEG recording presented for further evaluation. Since age 5, the patient had focal seizures that began with a “funny feeling,” flushing, and head turning to the right. Some seizures terminated at this point, while at other times her seizures progressed to impair awareness and/or evolved into a bilateral convulsive seizure. Treatment with several antiseizure medicines failed to improve the woman’s seizure frequency. An MRI did not demonstrate any structural brain abnormalities. An EMU evaluation captured seizures with a broad, right hemispheric onset. An initial MEG recording prior to VNS implantation captured epileptiform activity, which was originally and unsuccessfully analyzed using equivalent current dipole modeling (Figure 4A). The interpretation at the time was that the MEG study did not provide localizing information. As the patient’s seizure focus was not localized with any modality, she received a VNS.

However, the patient continued to have seizures after the VNS implantation. Because we had subsequently implemented SAM(g2) as an alternative to dipole analyses, the patient returned for a second MEG scan. SAM(g2) was used to analyze this second MEG recording, as well as the previous recording performed prior to VNS implantation. Even though the second recording was separated from the first by six years and the raw MEG sensor data were strongly contaminated by artifact from the VNS during the second recording, an equivalent right frontal focus was identified on both recordings (Figure 5A), a focus not revealed by dipole analysis previously. This indicates that SAM can reproducibly localize interictal epileptiform activity despite the presence of large artifacts due to the VNS implant. (The dipole localization in the presence of VNS artifact is shown in Figure 4B for comparison. The dipole tails have been omitted for clarity.) Examination of the virtual electrodes reconstructed from the SAM(g2) focus demonstrated MEG epileptiform discharges that correlated with the simultaneously recorded scalp EEG during the second recording (Figure 5C), and, importantly, that lacked the high-amplitude fluctuations present in the raw MEG data that were induced by the VNS (Figure 5B). (For comparison, the raw MEG sensor data prior to VNS implantation are depicted in the top part of Figure 5B.) Furthermore, during the patient’s second MEG recording an electrographic seizure was recorded on EEG, with a preceding MEG-recorded discharge detected in the virtual electrode (Figure 5D). Given the new localizing information provided by the MEG recordings, the patient was determined to be a candidate for ICEEG and possible resection. A subdural grid was placed over the right frontal lobe, covering the focus identified on MEG (Figure 5F). Seizures captured during the invasive monitoring demonstrated electrographic onset (Figure 5E) very close to the focus of peak kurtosis identified by SAM(g2) on the MEG recordings (Figure 5F). Following resection, which included the focus identified by SAM(g2), the patient experienced a significant improvement in her seizures, improving from four to six focal seizures with loss of awareness and sometimes evolution to bilateral convulsive seizures monthly to two to four focal seizures with retained awareness monthly. Her scalp EEG recordings demonstrated a greatly reduced frequency of interictal epileptiform activity.

### 3.5. Case 5: MEG-Only Seizure

A 31-year-old woman with spells preceded by visual symptoms of macropsia and micropsia and olfactory auras presented for further evaluation. Following the auras, the patient often exhibited right eye deviation and tonic flexion of the right upper extremities followed by a loss of awareness and generalized tonic-clonic events of 2–3 min in length. Afterwards, the patient experienced mild confusion and was tired for tens of minutes. She also had jerking of her arms and legs during sleep and the patient’s husband was unable to arouse her. The patient’s auras occurred every few days. She had 1–2 daytime spells a month, and she estimated that she had several more nighttime spells. These spells had been captured on routine EEG and during two EMU admissions and no electrographic change suggestive of seizure was noted. Neither a CT nor an MRI demonstrated any structural brain abnormalities. Treatment with several antiseizure medicines failed to provide adequate relief, and the patient had allergic reactions to some of the medicines. Despite the significant concern that her spells were nonepileptic in nature, given the severity of her events the patient received a diagnostic MEG scan. 

SAM(g2) identified three foci of epileptiform activity during her MEG scan: one focus in the left posterior temporal/lateral occipital cortex (Figure 6A), and bilateral mesial temporal foci (not shown). The left posterior temporal/lateral occipital cortical focus was likely responsible for her visual symptoms, and the bilateral mesial temporal foci were likely responsible for her olfactory auras as the irritable zones encompassed entorhinal cortex. The irritability of the mesial temporal cortices was initially supported by bilateral sphenoidal electrodes, which detected interictal epileptiform activity. Importantly, MEG captured two seizures during the recording which originated within the posterior temporal/lateral occipital cortex, and neither of these seizures was discernable on EEG (Figure 6B). Based on the MEG localizing information, ICEEG was performed as part of a presurgical evaluation. During a recorded seizure, ICEEG demonstrated early epileptiform activity (red channel, Figure 6D) at electrodes (green) near the SAM(g2) peaks (red crosses, Figure 6C). While ICEEG confirmed the localization of the three SAM(g2) foci, it also corroborated the MEG finding that the posterior temporal/lateral occipital focus was indeed the seizure initiator. Interestingly, the morphology of the ICEEG seizure was very similar to the morphology of the MEG seizure; both started with an initial spike, then exhibited a period of fast activity, and then evolved into a rhythmic discharge. Because the patient’s primary focus encompassed part of visual cortex, she decided against surgery to avoid a potential visual field defect. However, the MEG findings were still useful as they definitively identified the patient as having epilepsy. The patient is currently being considered for responsive neurostimulation device (RNS) placement.

## 4. MEG in Research

We have undertaken several clinical studies as part of our ongoing effort to improve patient care. These research studies are possible because of the advances and capabilities developed through use of MEG as a clinical diagnostic tool. As mentioned above, we have explored the uses of SAM to remove artifacts associated with dental hardware, with biological devices located outside the brain volume, such as those arising from VNS devices or pacemakers, or even artifacts arising from within the brain volume such as those associated with CSF shunts. It should be mentioned that alternative noise reduction techniques such as signal space separation [120] cannot accommodate noise arising from within the brain itself. The goal of such studies is to enable patients with devices to be the subject of research studies, allowing for advances to be made in this medically-complex population. We also have ongoing studies of new MEG language and other functional tests, as well as additional ways to map seizure networks or seizure foci with MEG/MSI.

In addition to our clinical research, the WFBH MEG lab maintains a vibrant basic and translational research program. In contrast to fMRI, MEG is a direct measure of brain activity, so use of MEG/MSI more accurately determines the neurophysiologic activity of brain networks, which underlies the processing of information. This approach is used to study cognitive processes in multiple pathological conditions including epilepsy, post-traumatic stress disorder, traumatic brain injury, and by drugs of abuse. These studies are performed both humans and nonhuman primates. The cognitive and precognitive tasks developed can be used to probe network behavior in each of these conditions, and are deployed as translational assays of function in populations as diverse as children, Veterans with posttraumatic stress disorder (PTSD) and/or traumatic brain injury (TBI) [117,118], normal adults, and nonhuman primates [119,121]. The goal in coupling MEG with probes of brain function is to develop a set of metrics that typify a particular disorder, and may eventually lead to better diagnosis and clinical interventions. These unique research capabilities allow for collaboration, which synergizes with clinical referrals between institutions.

## 5. Conclusions

MEG can be a highly utilized tool for both clinical and research use. Clinically, MEG can provide critical information in often difficult epilepsy cases, where other imaging modalities may be inconclusive. Our experience suggests that MEG can lead to demonstrably improved patient outcomes, particularly in the more difficult cases, as highlighted in the selected cases above. The ability to map eloquent cortex enhances clinical decision making in complex surgical cases. Our utilization of SAM provides the fine level of detail similar to the signals obtained by ICEEG, without the need for invasive procedures. Better utilization of MEG/MSI leads to much better localization and lateralization of seizure foci, more tailored ICEEG coverage, and increased utilization of surgery. Patients who receive surgery have a much better chance of epilepsy freedom or seizure reduction and experience improved quality of life, all of which reduce treatment and societal costs [25,34,37]. Research uses of the MEG have been very successful, by providing information on whole-brain activity at levels of resolution not possible by any other approach, both in humans and in nonhuman primates. Furthermore, the MEG represents a translational science success story, as the advanced analyses techniques for MEG and technical experience have led to improved patient care. As continuing advances in our understanding of the brain open up new opportunities for diagnosis and treatment of individuals in clinical settings, and for continued advancements in research settings, MEG is uniquely positioned to provide measures of whole-brain activity with a combination of high temporal and spatial resolution which are not available by any other means.

## Figures and Tables

**Figure 1 brainsci-08-00157-f001:**
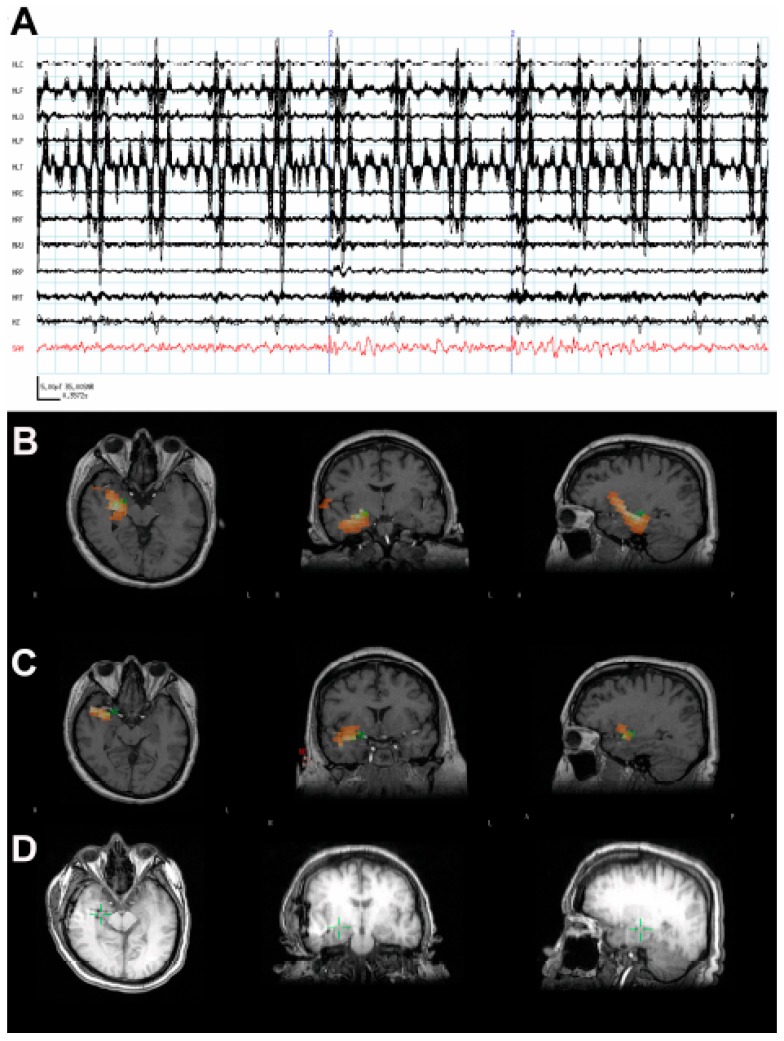
(**A**) The raw magnetoencephalogram (MEG) sensor data (black traces) exhibited strong artifacts due the patient’s vagal nerve stimulator (VNS), but the virtual electrode signal (red) from the amygdala displayed clear spikes. (**B**) The synthetic aperture magnetometry (SAM) (g2) statistical parametric maps indicated a right hippocampal focus as well as (**C**) another focus in the amygdala. (**D**) A computerized tomography (CT) scan reveals the placement of the hippocampal and amydalar depth electrodes, as well as the location of the hippocampal focus (green cross) as identified by SAM(g2). (**E**) An example of a seizure that arose from the anterior hippocampal and amygdalar electrodes, (black asterisks). (**F**) A coronal plane ictal single-photon emission computerized tomography (SPECT) image demonstrating hyperperfusion of the right temporal lobe (arrow). (**G**) A coronal plane 18F-fluorodeoxyglucose (FDG) (positron emission tomography) PET image demonstrating hypometabolism of the right temporal lobe (arrow).

**Figure 2 brainsci-08-00157-f002:**
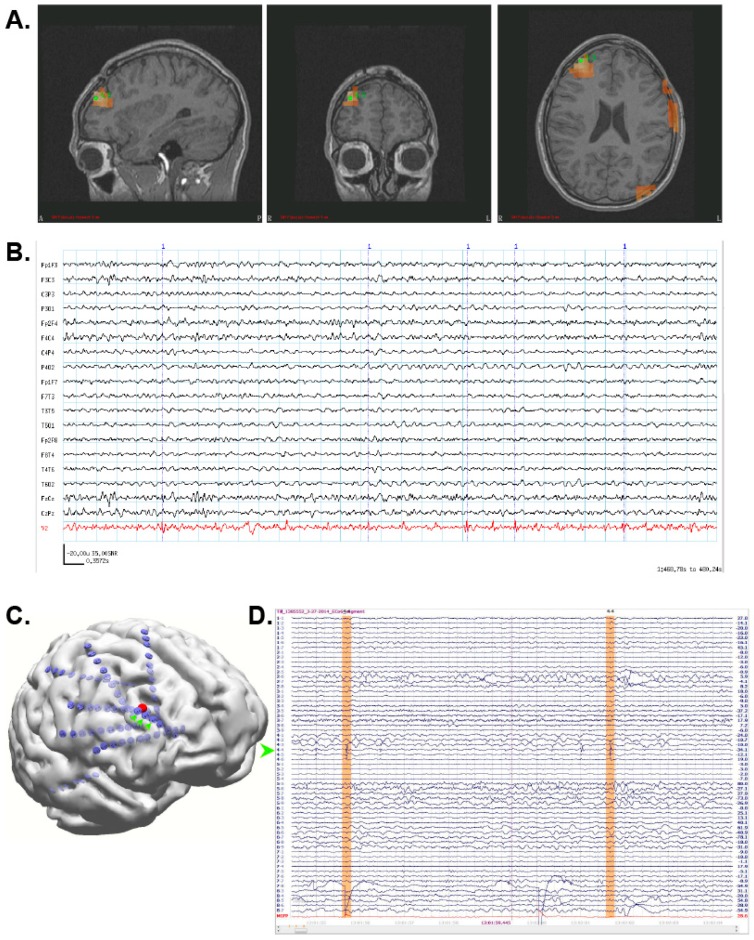
(**A**) SAM(g2) identified a single focus in the right middle frontal gyrus. (**B**) The virtual electrode (red) exhibited numerous small, sharp spikes which were not visible in the simultaneous EEG (black traces). (**C**) Stereo EEG electrodes (green arrows) placed directly over the MEG focus (red sphere) exhibited frequent epileptiform activity ((**D**), green arrow), two spikes of which are highlighted in orange.

**Figure 3 brainsci-08-00157-f003:**
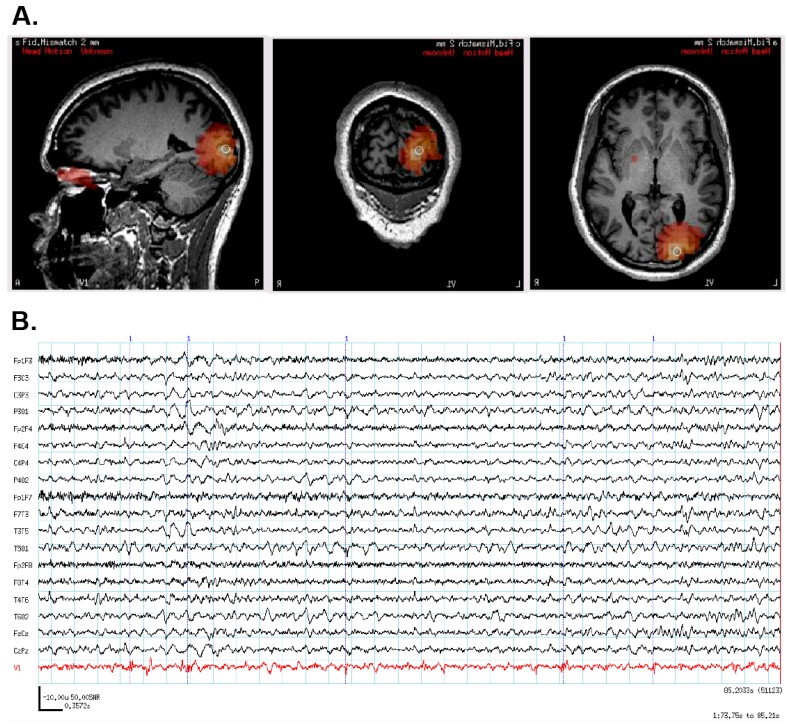
(**A**) SAM(g2) indicated the presence of a left occipital focus with numerous MEG-only spikes visible on the virtual electrode ((**B**), red trace). The simultaneous scalp EEG recording (black traces) did not detect these spikes.

**Figure 4 brainsci-08-00157-f004:**
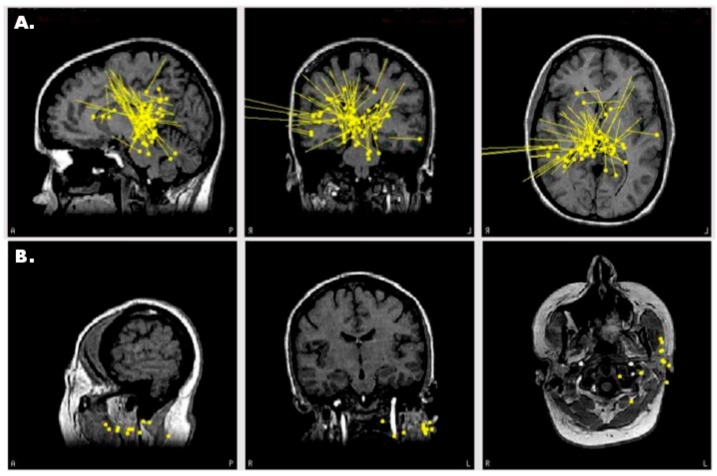
(**A**) Prior to VNS implantation, equivalent current dipole map of interictal spikes was nonlocalizing. The yellow circle indicates the dipole position, and the yellow tail indicates dipole magnitude and orientation. (**B**) Dipole map of interictal spikes following VNS implantation. Dipoles localize to VNS leads in the patient’s neck. Tails have been omitted for clarity. All images are in radiological coordinates, where the patient’s left is presented on the right side of each figure.

**Figure 5 brainsci-08-00157-f005:**
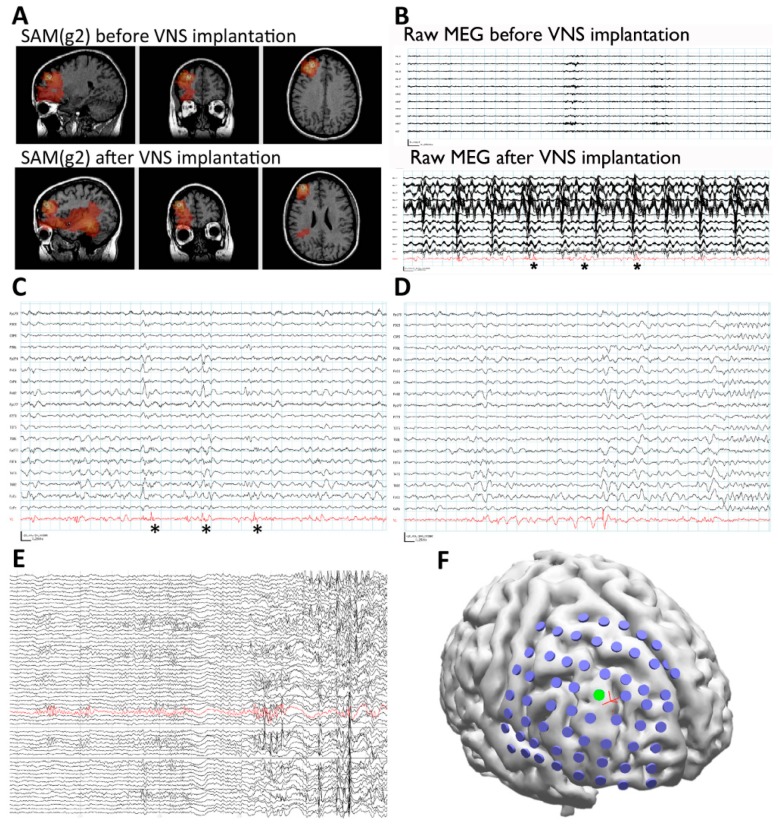
(**A**) SAM(g2) analysis of the MEG recording before VNS implantation (upper panel) and after VNS implantation (lower panel) identifies a peak of kurtosis at the same anatomical location. (**B**) The raw MEG recording (black traces, displayed as a butterfly plot) before VNS implantation (upper panel) and after VNS implantation (lower panel). The raw MEG recording after VNS implantation was heavily contaminated by artifact from the patient’s VNS, while the virtual electrode signals (lower panel, red trace) permitted the identification of epileptiform activity (lower panel, asterisks). (**C**) Epileptiform activity (asterisks) identified within the virtual electrode (red trace) coincided with poorly localized activity on the simultaneously recorded scalp EEG (black traces). (**D**) A run of epileptiform activity was seen in the virtual electrode (red trace) prior to a poorly localized discharge that was observed on scalp EEG (black traces). (**E**) Electrocorticography was used to identify an ictal focus (red trace). (**F**) Reconstruction of the patient’s brain from her own MRI illustrates the placement of the subdural grid (blue disks). The electrode that was determined to overlie the ictal focus (green disk, red trace from (**E**)) colocalized with the peak identified from the MEG recording (red cross).

**Figure 6 brainsci-08-00157-f006:**
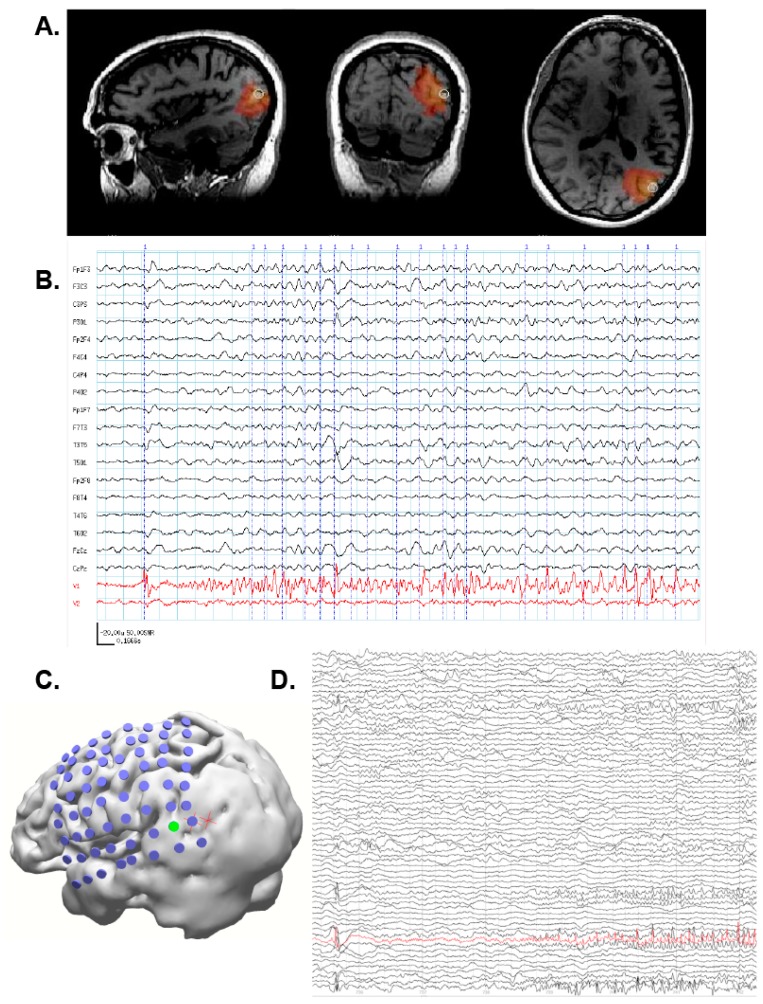
(**A**) SAM(g2) maps reveal a strong focus in the left posterior temporal/lateral occipital cortex. (**B**) A MEG-only electrographic seizure emanating from the left temporal/occipital focus was visible in the upper virtual electrode (red, top trace). The patient reported experiencing her aura during this event, which later evolved into a tonic clonic event and necessitated her removal from the scanner. The lower virtual electrode (red, bottom trace) corresponds to activity from one of the mesial temporal foci, but epileptiform signals were not evident during this initial time frame. The simultaneous EEG (black, upper traces) did not reveal the discharge. (**C**) During a recorded seizure, electrocorticography (ECoG) demonstrates early epileptiform activity (red channel) at an electrode ((**D**), green) near the SAM(g2) peaks (red crosses). The ECoG positions are approximate because they were reconstructed based on x-ray images and because the patient experienced swelling after implantation.

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
