# Peer review of "Magnetoencephalography: Clinical and Research Practices"

_brainsci, 2018, doi:10.3390/brainsci8080157_

Reviewer 1 Report

This is a review article that summarizes the research efforts and clinical studies performed at Wake Forest School of Medicine. The manuscript is in general well-written and covers a broad range of the available literature. Although the manuscript provides a fair review of the research performed at Wake Forest School of Medicine, it misses a section dedicated to the MEG research from other laboratories. This section can be listed as Section 1.2. before the “Clinical Uses of MEG” and be entitled as “Use of MEG in basic and translational research”. I  fully understand that it is impossible to summarize the use of MEG in all research but at least an attempt to cover the main topics of basic and translational research should be included and will be appreciated by the readers.

For example, the literature on the use of MEG for understanding the different systems in the brain can be reviewed and summarized here [i.e. for somatosensory see: Hoshiyama et al., 1997; Nakamura et al., 1998; Inui et al., 2004; Papadelis et al., 2011; 2012; Ioannides et al., 2013), for motor see: Cheyne et al., 2008; Cheyne et al., 2013; for visual: Shigihara and Zeki 2014, etc). Moreover, there are inaccurate statements in the manuscript that should be corrected (see detailed comments below), and available literature that is missing that should be discussed and cited (see detailed comments below).

Line 19: MEG accurately localizes seizure foci. This statement is inaccurate. MEG is mostly used for the detection of interictal epileptic discharges and thus the localization of the irritative zone, not the seizure foci (or seizure onset). The detection and localization of the seizure onset is possible with MEG but much more difficult (compared to EEG) since MEG recordings usually do not last more than an hour and in most patients seizures occur only few times per day or even per week. Please rephrase.

Line 14: SAM is one of the many source localization technique used to solve the inverse problem. I do not see a reason to highlight its use in the second sentence of the abstract.

Line 45: Unlike to EEG, which is restricted to cortical signals, MEG can also be used to image subcortical brain activity… This statement is not correct. In principle, EEG detects sources with both tangential and radial orientation. Thus, it is more capable to detect sources which are located in deep structures compared to MEG. Subcortical sources which are close to the center of the sphere generate almost zero magnetic field that is measurable outside the scalp. Please rephrase.

Line 45: The review of the existing literature is incomplete. Please cite Balderston et al. 2013; Dumas et al., 2013; and Styliadis et al. 2014 for the localization of amygdala.  

Line 47: One of the major abilities of MEG (compared to EEG) is the localization of activity from the cerebellum. Please review and cite: Martin et al., 2006 and Styliadis et al., 2015.

Line 53: How come the first figure appears in the manuscript in the number 4?

Line 63: or even EEG (due to conductivity of the head)…. This is a wrong statement. In contrast to what few scientists believe the conductivity of the head does not have any effect in the frequency content of the brain signals. EEG and MEG are able to detect the same frequency bandwidths.

Line 64: You may want to review and summarize briefly at this point the ability of MEG to detect and localize high frequency oscillations (please cite: Leiken et al., 2014; Papadelis et al., 2009).

Line 69: Please add here and cite Styliadis et al., 2015 NeuroImage for the use of virtual channels in the cerebellum.

Line 85: As the author correctly stated, the use of epilepsy surgery is more critical in children. Please discuss further and expand this section here. Please consider citing the following articles: Papadelis et al., 2013 Current and emerging potential for MEG in pediatric epilepsy. Gaetz et al., 2015. Magnetoencephalography for Clinical Pediatrics: Recent Advances in Hardware, Methods, and Clinical Applications.

Line 99: This paragraph emphasizes the use of MEG for the detection of the seizure onset zone (SOZ). Although MEG is indeed able to detect and localize the SOZ in some cases, it is mostly used for the detection of interictal epileptic discharges and the localization of the irritative zone. Please change your writing in this paragraph. Emphasize the use of MEG in the detection of spikes first and state its sensitivity and specificity.  Then, discuss the localization of the SOZ.

Line 131: I would strongly suggest to dedicate a paragraph in the use of MEG for the detection and localization of HFOs. There are many papers now showing that MEG is able to non-invasively detect and localize HFOs (see Tamilia et al., 2017 Frontiers in Neurology; Papadelis et al., JoVE 2016; van Klink et al., 2016 Clinical Neurophysiology).

Line 107: For the localization of spikes with MEG please consider citing Hunold et al., Front Hum Neuroscience 2014 and Jansen et al., 2006.

Line 178: I would suggest to dedicate one figure to common MEG artifacts that often harm good quality MEG recordings.

Author Response

Comments and Suggestions for Authors

This is a review article that summarizes the research efforts and clinical studies performed at Wake Forest School of Medicine. The manuscript is in general well-written and covers a broad range of the available literature. Although the manuscript provides a fair review of the research performed at Wake Forest School of Medicine, it misses a section dedicated to the MEG research from other laboratories. This section can be listed as Section 1.2. before the “Clinical Uses of MEG” and be entitled as “Use of MEG in basic and translational research”. I  fully understand that it is impossible to summarize the use of MEG in all research but at least an attempt to cover the main topics of basic and translational research should be included and will be appreciated by the readers.

For example, the literature on the use of MEG for understanding the different systems in the brain can be reviewed and summarized here [i.e. for somatosensory see: Hoshiyama et al., 1997; Nakamura et al., 1998; Inui et al., 2004; Papadelis et al., 2011; 2012; Ioannides et al., 2013), for motor see: Cheyne et al., 2008; Cheyne et al., 2013; for visual: Shigihara and Zeki 2014, etc). Moreover, there are inaccurate statements in the manuscript that should be corrected (see detailed comments below), and available literature that is missing that should be discussed and cited (see detailed comments below)

We greatly thank the reviewer for these comments and we have added all of the suggested references. Our primary focus in this paper is the use of MEG in the treatment of epilepsy and we are indebted to all of the research teams that have pioneered many of the studies that we discuss in the manuscript, both in terms of the work on epilepsy and also in the techniques related to functional mapping.

 Line 19: MEG accurately localizes seizure foci. This statement is inaccurate. MEG is mostly used for the detection of interictal epileptic discharges and thus the localization of the irritative zone, not the seizure foci (or seizure onset). The detection and localization of the seizure onset is possible with MEG but much more difficult (compared to EEG) since MEG recordings usually do not last more than an hour and in most patients seizures occur only few times per day or even per week. Please rephrase.

 We appreciate this point and have attempted to clarify our ambiguous language throughout the manuscript.

 Line 14: SAM is one of the many source localization technique used to solve the inverse problem. I do not see a reason to highlight its use in the second sentence of the abstract.

 There are many useful localization methods, but beamforming is less commonly used for clinical purposes and we hoped to modestly add to this discussion. In particular, the ability of SAM(g2) to image kurtotic spikes is especially useful for epilepsy, and we feel this adds significance to the paper, as well as being the main analysis pipeline with which we have experience.

 Line 45: Unlike to EEG, which is restricted to cortical signals, MEG can also be used to image subcortical brain activity… This statement is not correct. In principle, EEG detects sources with both tangential and radial orientation. Thus, it is more capable to detect sources which are located in deep structures compared to MEG. Subcortical sources which are close to the center of the sphere generate almost zero magnetic field that is measurable outside the scalp. Please rephrase.

 We have clarified our language to reflect this point.

 Line 45: The review of the existing literature is incomplete. Please cite Balderston et al. 2013; Dumas et al., 2013; and xStyliadis et al. 2014 for the localization of amygdala.  

 We have added these citations.

 Line 47: One of the major abilities of MEG (compared to EEG) is the localization of activity from the cerebellum. Please review and cite: Martin et al., 2006 and Styliadis et al., 2015.

 We have added these references.

 Line 53: How come the first figure appears in the manuscript in the number 4?

 This figure is part of a sequence of cases but it had relevance to the description in this paragraph and we hoped that it would provide an illustration of the concept.

 Line 63: or even EEG (due to conductivity of the head)…. This is a wrong statement. In contrast to what few scientists believe the conductivity of the head does not have any effect in the frequency content of the brain signals. EEG and MEG are able to detect the same frequency bandwidths.

 We have clarified our language to reflect this point.

 Line 64: You may want to review and summarize briefly at this point the ability of MEG to detect and localize high frequency oscillations (please cite: Leiken et al., 2014; Papadelis et al., 2009).

 We have included these references in our new section on HFOs and MEG.

 Line 69: Please add here and cite Styliadis et al., 2015 NeuroImage for the use of virtual channels in the cerebellum.

 We have added these references.

 Line 85: As the author correctly stated, the use of epilepsy surgery is more critical in children. Please discuss further and expand this section here. Please consider citing the following articles: Papadelis et al., 2013 Current and emerging potential for MEG in pediatric epilepsy. Gaetz et al., 2015. Magnetoencephalography for Clinical Pediatrics: Recent Advances in Hardware, Methods, and Clinical Applications.

 Surgery for pediatric patients with epilepsy is certainly important and also underutilized. We have added a new paragraph on MEG in pediatric patients and we have included the above references.

 Line 99: This paragraph emphasizes the use of MEG for the detection of the seizure onset zone (SOZ). Although MEG is indeed able to detect and localize the SOZ in some cases, it is mostly used for the detection of interictal epileptic discharges and the localization of the irritative zone. Please change your writing in this paragraph. Emphasize the use of MEG in the detection of spikes first and state its sensitivity and specificity.  Then, discuss the localization of the SOZ.

 We appreciate this point and have attempted to clarify our ambiguous language throughout the manuscript.

 Line 131: I would strongly suggest to dedicate a paragraph in the use of MEG for the detection and localization of HFOs. There are many papers now showing that MEG is able to non-invasively detect and localize HFOs (see Tamilia et al., 2017 Frontiers in Neurology; Papadelis et al., JoVE 2016; van Klink et al., 2016 Clinical Neurophysiology).

 We have added a new section on the importance of HFOs and have included the above, useful references.

 Line 107: For the localization of spikes with MEG please consider citing Hunold et al., Front Hum Neuroscience 2014 and Jansen et al., 2006.

             We have added these references.

 Line 178: I would suggest to dedicate one figure to common MEG artifacts that often harm good quality MEG recordings.

Artifacts are certainly an interesting and challenging aspect of MEG analysis, and there are many ways in which different groups have attempted to mitigate them. We tried to address that aspect by indicating that others have used tSSS and SAM in a variety of circumstances for artifact removal, and we have addressed that also in a separate publication. We hope that figures 1 and 5 present some indication of the types of noise that SAM can minimize as well.

Reviewer 2 Report

This paper gives an overview of clinical magnetoencephalography (MEG), which records neuromagnetic fields of the brain.  The authors introduce their clinical practice at Wake Forest  Baptist Health, using both conventional single dipole analysis and synthetic aperture magnetometry (SAM).  They also present five patients with intractable epilepsy who underwent MEG and subsequent intracranial EEG (IEEG) recording.  The MEG results, especially the SAM analysis, demonstrate localization of spike sources that are consistent with IEEG.  Here are some comments:

Provide a short description of SAM analysis, especially how to obtain the waveforms at virtual sensors.

Describe shortly how to analyze epileptic spikes.  Are they visually recognized?  How many?  What part of the spikes was analyzed (i.e., peak, onset, etc?).

Provide details of the data acquisition, including sampling frequency and filtering.  Also the type of MRI sequences, such as MPRAGE, and preprocessing, if any.

Spell out CSF, RNS, SPECT and PET.  Although the words vagus nerve stimulator, post-traumatic stress disorder and traumatic brain injury appear in the text, they should be clearly stated to indicate 'VNS', 'PTSD' and 'TBI'.

Author Response

This paper gives an overview of clinical magnetoencephalography (MEG), which records neuromagnetic fields of the brain.  The authors introduce their clinical practice at Wake Forest  Baptist Health, using both conventional single dipole analysis and synthetic aperture magnetometry (SAM).  They also present five patients with intractable epilepsy who underwent MEG and subsequent intracranial EEG (IEEG) recording.  The MEG results, especially the SAM analysis, demonstrate localization of spike sources that are consistent with IEEG.  Here are some comments:

 Provide a short description of SAM analysis, especially how to obtain the waveforms at virtual sensors.

Describe shortly how to analyze epileptic spikes.  Are they visually recognized?  How many?  What part of the spikes was analyzed (i.e., peak, onset, etc?).

 We have attempted to briefly describe the SAM(g2) process as well as how it is integrated with EEG.

 Provide details of the data acquisition, including sampling frequency and filtering.  Also the type of MRI sequences, such as MPRAGE, and preprocessing, if any.

 We have added a more detailed section on MEG preprocessing and we have also detailed which MRI sequences we use.

Spell out CSF, RNS, SPECT and PET.  Although the words vagus nerve stimulator, post-traumatic stress disorder and traumatic brain injury appear in the text, they should be clearly stated to indicate ‘VNS’, ‘PTSD’ and ‘TBI’.  xREFS

 All abbreviations are now elucidated, including our newly added MRI sequences.

Round  2

Reviewer 1 Report

The authors worked extensively in the manuscript and satisfied all my concerns.